# The Role and Expression of Angiogenesis-Related miRNAs in Gastric Cancer

**DOI:** 10.3390/biology10020146

**Published:** 2021-02-12

**Authors:** Martina Giuppi, Anna La Salvia, Jessica Evangelista, Michele Ghidini

**Affiliations:** 1Faculty of Medicine, CEU San Pablo University, 28003 Madrid, Spain; mar.giuppi@ceindo.ceu.es; 2Department of Oncology, University Hospital 12 de Octubre, 28041 Madrid, Spain; alasalvi@ucm.es; 3Thoracic Surgery, Fondazione Policlinico Universitario A. Gemelli IRCCS, 00168 Rome, Italy; evangelistajessica664@gmail.com; 4Oncology Unit, Fondazione IRCCS Ca’ Granda Ospedale Maggiore Policlinico, 20122 Milan, Italy

**Keywords:** Micro RNAs (miRNAs), gastric cancer, angiogenesis, VEGF, ramucirumab, biomarkers

## Abstract

**Simple Summary:**

The aberrant expression of several micro RNAs (miRNAs) has been shown to be involved in neoplastic angiogenesis, which is a crucial mechanism in gastric cancer onset and progression. In this review, the possible prognostic and predictive roles of angiogenesis-related miRNAs as novel biomarkers of gastric cancer have been evaluated, but neither tissue nor circulating biomarkers have shown a predictive role for response to anti-angiogenic treatment. Nevertheless, we consider that in future studies, miRNAs should be evaluated as candidate biomarkers with prognostic and predictive features.

**Abstract:**

Gastric cancer (GC) is the fifth most frequently diagnosed malignant tumor and the third highest cause of cancer mortality worldwide. For advanced GC, many novel drugs and combinations have been tested, but results are still disappointing, and the disease is incurable in the majority of cases. In this regard, it is critical to investigate the molecular mechanisms underlying GC development. Angiogenesis is one of the hallmarks of cancer with a fundamental role in GC growth and progression. Ramucirumab, a monoclonal antibody that binds to vascular endothelial growth factor-2 (VEGFR-2), is approved in the treatment of advanced and pretreated GC. However, no predictive biomarkers for ramucirumab have been identified so far. Micro RNAs (miRNAs) are a class of evolutionarily-conserved single-stranded non-coding RNAs that play an important role (via post-transcriptional regulation) in essentially all biologic processes, such as cell proliferation, differentiation, apoptosis, survival, invasion, and migration. In our review, we aimed to analyze the available data on the role of angiogenesis-related miRNAs in GC.

## 1. Introduction

### 1.1. Angiogenesis and Cancer

Angiogenesis is a key player in the growth of cancer [1]. Tumor-associated neo-vasculature is important for delivering nourishment and oxygen to growing tumors. In addition, angiogenesis partakes in multiple aspects of tumor biology, including dissemination and metastasis processes [2], metabolic deregulation [3], and cancer stem cell maintenance [4,5]. Angiogenesis is a complex process by which new blood vessels are formed from pre-existing ones [6]. In normal conditions, the vasculature becomes largely quiescent. However, within tumors, an “angiogenic switch” is always activated, leading to the continuous generation of new vessels [7]. This “angiogenic switch” is governed by the upregulation of pro-angiogenic and downregulation of anti-angiogenic signals produced by tumor cells or the tumor microenvironment [8,9,10].

### 1.2. Angiogenesis and Gastric Cancer

The vascular endothelial growth factor (VEGF) family is a crucial mediator of angiogenesis [11]. The VEGF family consists of seven major subtypes, including VEGF-A, VEGF-B, VEGF-C, VEGF-D, VEGF-E, and placental grow factor 1 and 2 (PIGF-1 and PIGF-2, respectively) [12]. VEGF subtypes stimulate cellular responses by binding to tyrosine kinase receptors (VEGFR-1, -2, -3) on the cell surface. VEGF is produced by several cell types, such as fibroblasts, inflammatory cells and many tumoral cells, often in response to increasing tumor hypoxia via the hypoxia inducible factor-1α (HIF-1α) pathway. Notably, the activation of VEGFR has a critical role in GC angiogenesis [13]. Approximately 50% of GCs express VEGF, and the overexpression of VEGF-A and VEGF-D in GC is associated with poor prognosis [11,14]. The hypoxia inducible factor (HIF) pathway is also involved in angiogenesis. HIF-1 is a dimeric protein (HIF-1α, HIF-1β) complex that plays an important role in the response to low oxygen concentrations or hypoxia. HIF-1 is a crucial physiological regulator of homeostasis, vascularization, and anerobic metabolism, but in cancer, it allows for the survival and proliferation of cancerous cells due to its angiogenic properties. HIF-1 is responsible for the migration of mature endothelial cells towards a hypoxic environment via the regulation of VEGF transcription [15].

A third important pathway for angiogenesis and tumorigenesis involves the hepatocyte growth factor (HGF)/tyrosine-protein kinase MET (c-MET) signaling pathway. HGF is a pleiotropic cytokine that has been reported to prevent and attenuate disease progression by influencing multiple pathophysiological processes and promoting cell proliferation, survival, motility, scattering, differentiation, and morphogenesis [16,17]. c-MET is a receptor tyrosine kinase that binds with its ligand (HGF) and activates a wide range of different cellular signaling pathways, including those that are involved in proliferation, motility, migration and invasion [17]. When the tyrosines within the multifunctional docking site become phosphorylated, they recruit: (1) signaling effectors such as the adaptor proteins growth factor receptor-bound protein 2 (GRB2), src homology 2 domain-containing (SHC), v-crk sarcoma virus CT10 oncogene homolog (CRK) and CRK-like (CRKL); (2) effector molecules such as phosphoinositide 3-kinase PI3K, phospholipase Cγ (PLCγ), the proto-oncogene tyrosine-protein kinase SRC, and the src homology 2 domain-containing 5′ inositol phosphatase (SHIP-2); and (3) signal transducer and activator of transcription-3 (STAT-3) [18,19].

The phosphatidylinositol 3-kinase (PI3K) signaling pathway regulates growth, survival, proliferation, and angiogenesis [20,21,22]. This pathway has two major positive and negative regulators, PI3K and phosphatase and tensin homolog (PTEN), respectively, which are two of the most frequently mutated proteins in human cancers that are involved in tumorigenesis. PTEN mainly regulates PI3K signaling by dephosphorylating the lipid signaling intermediate, phosphatidylinositol (3,4,5)-trisphosphate (PIP_3_), but may have additional phosphate-independent activities and other functions in the nucleus. PTEN is a tumor suppressor gene that negatively regulates mammalian target of rapamycin (mTOR) complex 1 (mTORC1) activity, which activates the translation of proteins. The restoration of PTEN expression may block angiogenesis in GC by inactivating the PI3K/protein kinase B (AKT) pathway [23]. Additionally, several miRNAs have been identified to target the Forkhead box O (FOXO) transcription factors and the tuberous sclerosis complex subunit 1 (TSC1) [23]. FOXO is a key substrate of AKT. AKT-mediated phosphorylation of the transcription factor FOXO can increase proliferation and survival. TSC1 and TSC2 form a complex that inhibits the activity of the small G protein Rheb. AKT-mediated phosphorylation of TSC2 relieves its inhibition of Rheb activity, leading to the activation of mTORC1 [22]. Nuclear factor kappa B (NF-κB) functions as a dimeric transcription factor that regulates the expression of genes influencing a broad range of biological processes including innate and adaptive immunity, inflammation, stress responses, B-cell development, and lymphoid organogenesis [24].

Signal transducer and activator of transcription-3 (STAT-3) is a member of a family of seven proteins (STATs 1, 2, 3, 4, 5a, 5b, and 6) that relay signals from activated cytokine and growth factor receptors in the plasma membrane to the nucleus, where they regulate gene transcription. STAT-3 activated genes block apoptosis, favor cell proliferation and survival, promote angiogenesis and metastasis, and inhibit antitumor immune responses [25]. Among these seven STAT proteins, STAT-3 plays a critical role in angiogenesis [26,27].

Many clinical trials have demonstrated that GC patients can benefit from angiogenesis inhibitors [28,29,30]. Ramucirumab is a monoclonal antibody that selectively binds to VEGFR-2, blocking the downstream effects of the VEGF pathway in angiogenesis. The survival benefits in the REGARD [31] and RAINBOW [32] studies led to the approval of ramucirumab for the treatment of advanced GC (hazard ratio (HR) for overall survival (OS): 0.776, 95% confidence interval (CI): 0.603–0.998, *p* = 0.047; HR for OS: 0.807, 95% CI: 0.678–0.962, *p* < 0.0001, respectively; and HR for progression-free survival (PFS): 0.483, 95% CI: 0.376–0.620, *p* < 0.0001, HR for PFS: 0.635, 95% CI: 0.536–0.752, *p* < 0.0001, respectively). Recently, there has been increasing interest in the development of new anti-angiogenic options for GC. Apatinib, a tyrosine kinase inhibitor (TKI) targeting VEGFR-2, was tested in a phase III placebo-controlled trial in Chinese patients with advanced pretreated GC. Rivoceranib (apatinib) treatment significantly improved PFS and OS with compared to placebo, but with a poor clinical impact on survival outcomes (median PFS: 2.6 for apatinib vs. 1.8 months for placebo, *p* < 0.001; median OS: 6.5 for apatinib vs. 4.7 months for placebo, *p* < 0.0149) [33]. In the randomized phase III placebo-controlled ANGEL trial, including both Eastern and Western patients, rivoceranib treatment significantly improved median OS from fourth line treatment (6.43 for apatinib vs. 4.73 months for plaebo, *p* = 0.0195), while median PFS was improved from third line treatment (2.83 for apatinib vs. 1.77 months for placebo, *p* < 0.0001) [34]. Among ongoing and recruiting studies, a phase II trial is evaluating the combination of apatinib and the programmed death-1 (PD-1) inhibitor sintilimab in unresectable GC with oligometastases as conversion therapy (NCT04267549). In another trial, apatinib is being combined with docetaxel and S-1 as first line treatment for advanced GC (NCT03154983). Moreover, the TKI regorafenib is being tested as maintenance treatment after first-line treatment for metastatic GC and absence of progression (NCT03627728). Lastly, a phase I/II trial is testing ramucirumab together with the poly ADP-ribose polymerase (PARP) inhibitor olaparib in advanced and unresectable GC (NCT03008278), while a phase II study is comparing ramucirumab plus 5-fluorouracil and irinotecan (FOLFIRI) to ramucirumab plus standard paclitaxel in second-line treatment (NCT03081143).

Although angiogenesis inhibitors can improve OS and PFS and achieve a better response rate in advanced GC, the clinical effect is quite different in individuals due to tumor heterogeneity. Therefore, it is essential to develop biomarkers that allow for identifying the subgroup of GC patients who would benefit from this type of therapy. These biomarkers could be used to predict efficacy and choose the most suitable patients to reduce unnecessary toxicity while maximizing clinical benefit [35].

### 1.3. MicroRNAs and Cancer

MicroRNAs (miRNAs) are a class of evolutionally conserved single-stranded noncoding RNAs of 19–22 nucleotides in length [36]. MiRNAs play an important role (via post-transcriptional regulation) in essentially all biologic processes such as cell proliferation, differentiation, apoptosis, survival, invasion, and migration [37]. MiRNAs form the RNA-inducing silencing complex (RISC)–miRNA functional unit, which regulates the expression of nearly 30% of known human genes [38] (Figure 1).

Nowadays, several detection methods are available for miRNAs. Among conventional techniques, Northern blotting analysis consists of the separation of RNA from the sample through denaturing gel electrophoresis with subsequent hybridization using a nucleic acid probe complementary to the target RNA. Differently, in situ hybridization uses labeled complementary nucleic acid probes to detect RNA in tissue sections or fixed cells. Similarly, microarrays rely on hybridization between target molecules and complementary probes. Reverse transcription (RT) quantitative polymerase chain reaction (qPCR) consists of two different steps: firstly, complementary DNA (cDNA) is synthetized using RT, secondly the product is amplified and detected using either intercalating dye or TaqMan probes [39]. The most innovative techniques include next-generation sequencing (NGS) methods and digital-droplet PCR (ddPCR). NGS is characterized by high data throughput. NGS uses cDNA from RNA reverse transcription. DNA is amplified many times in parallel and then sequenced multiple times simultaneously [40]. The ddPCR system has shown superior precision and sensitivity with respect to qPCR, being able to detect very low concentrations of target nucleic acids [41]. This technique is based on the partitioning of the nucleic acid sample into thousands of oil dispersed nanodroplets containing nucleic acid. Later, PCR amplification is carried out within each droplet [42]. Three basic mechanisms of miRNA-mediated gene regulation are known: translation repression, direct mRNA degradation, and miRNA-mediated mRNA decay [43]. Many studies have demonstrated that mutations in miRNA-encoding genes or the deregulated expression of miRNAs are integral to many human diseases, including cancer development and metastasis [44]. Thus, they can act as oncogenes or tumor suppressors depending on the function of their target genes. As a consequence, microRNAs have shown great promise for use in anti-metastatic cancer therapy.

### 1.4. MicroRNAs and Angiogenesis

Several miRNAs have been shown to be involved in neoplastic angiogenesis. In particular, VEGF expression in different types of cancer has been recognized to be regulated by different miRNAs, such as miR-20 [45], miR-29b [46], miR-93 [47], miR-126 [48], miR-190 [49], miR-195 [50], miR-200 [51], miR-203 [52], miR-497 [53], miR-503 [54], and miR-638 [55]. Some of these, such as miR-29, inhibit angiogenesis by downregulating VEGF when it is overexpressed. Others, such as miRNA-195, promote angiogenesis and metastasis via VEGF and pro-metastatic factors. Apart from directly targeting VEGF, a handful of miRNAs regulate VEGF-dependent tumor angiogenesis by targeting VEGF inducers, such as the HIF-1 pathway (miR-22 [56], miR-107 [57], miR-519c [58], miR-145 [59]). However, a direct connection between the role of miRNAs in angiogenesis and cancer metastasis remains to be established.

In this review, we will summarize the current evidence, provide new insights and discuss the main challenges around angiogenesis-related miRNAs in GC. Furthermore, our study will analyze the diagnostic and prognostic role of angiogenesis-related miRNAs as novel biomarkers of GC, and potential novel GC treatments based on miRNAs that have resulted from better molecular knowledge.

## 2. Materials and Methods

We performed a comprehensive literature review of the PubMed, Scopus, and Google Scholar databases regarding angiogenesis-related miRNAs in GC, using the terms “MiRNA” AND “angiogenesis” AND (“gastric cancer” OR “gastric carcinoma” or “gastric adenocarcinoma”). The search was limited to articles published in English. We considered all the original studies regarding angiogenesis-related miRNAs in GC. A total of 75 publications were identified, from which we selected a final pool of 28 articles based on their relevance in this context.

## 3. Results

Our search identified 28 studies focused on miRNAs targeting angiogenesis in GC. The details of the selected studies are reported in Table 1. We decided to order and divide the selected miRNAs according to their referral pathway. Therefore, we identified five groups: (1) miRNAs related to the VEGF pathway, (2) miRNAs involved in the HIF pathway, (3) miRNAs related to HGF/c-MET signaling, (4) miRNAs involved in the PI3K pathway, and (5) miRNAs related to STAT-3 signaling (Table 1 and Figure 2). We will briefly describe the importance in the angiogenesis process of these five pathways, and we will detail, for each of them, the significant studies about angiogenesis-related miRNAs that we have found through our literature search. Some of the analyzed studies focused on different miRNAs, so we have reported the same work in the different paragraphs where indicated.

### 3.1. MicroRNAs Involved in the VEGF Pathway

In this context, we have identified eight relevant published papers about miRNAs targeting VEGF signaling.

Wu et al. aimed to determine the potential relevance of miR-616-3p, located on the chromosome region 12q13.3, in GC tumorigenesis. In this study, miR-616-3p was demonstrated to facilitate tumor angiogenesis by elevating the expression level of VEGF-A/VEGFR2 in GC [60]. A second study suggested that the downregulation of another miRNA, miR-1, could favor angiogenesis in GC. In this analysis, miR-1 was under-expressed in primary GC tissues when compared with adjacent normal mucosa and several GC cell lines. The restoration of miR-1 significantly inhibited endothelial cell tube formation by decreasing the expression of VEGF-A and endothelin 1 (EDN1), the main angiogenic factors contributing to the development and maintenance of blood vessels [23]. Additionally, the overexpression of miR-126 led to direct inhibition of VEGF-A expression, reduction of cell proliferation, and inhibition of the angiogenic process, thereby inhibiting tumor growth both in vitro and in vivo in GC [61,62,63]. In another work, Zhang et al. reported that the downregulation of miR-29a/c increases VEGF expression and release in GC cells, with subsequent growth promotion of endothelial cells. MiR-29a/c significantly suppressed VEGF expression in GC cells, inhibiting cell growth and metastasis. By using a tumor mouse model, it was shown that secreted microvessels (MV) overexpressing miR-29a/c significantly reduced the growth of vasculature and tumors in vivo [64].

In addition, VEGF-C was found to be a direct target gene of three other miRNAs, miR-27b, miR-101, and miR-128 [65]. The expression levels of these miRNAs were inversely correlated with higher MV density. Thus, the overexpression of miR-27b, miR-101, or miR-128 was demonstrated to suppress migration, proliferation, and tube formation in human umbilical vascular endothelial cells (HUVECs) by repressing VEGF-C secretion in GC cells. MiR-27b, miR-101 and miR-128 inhibit angiogenesis by downregulating VEGF-C expression in GC [65]. Mei et al. reported that miR-590 can simultaneously regulate VEGFR1/2 in GC. In this study, miR-590 has also been shown to contribute to the regulation of the expression of neuropilin 1 (NRP1) in GC. NRP1 is a transmembrane protein that can bind the VEGF_165_ isoform and enhance cell migration via VEGFR2, also inducing vascular permeability and arteriogenesis. The authors reported that miR-590 was downregulated in GC tissues and cell lines, and this was related to the dysregulation of the transcription factor SNAIL. SNAIL inhibited the expression of miR-590, thereby upregulating the expression levels of NRP1 and VEGFR1/2; this led to the promotion of the epithelial–mesenchymal transition (EMT) process in GC and the upregulation of SNAIL. In addition, the overexpression of miR-590 inhibited the migration, invasion, proliferation, and digital microvascular (D-MVA) levels of gastric cancer cells in vivo and in vitro by targeting VEGFR1/2 and NRP1 [66].

### 3.2. MicroRNAs Involved in the HIF Pathway

In our search, we identified two studies reporting data about the aberrant expression in GC of HIF-related miRNAs. The first study by Zhang et al. demonstrated how under hypoxic conditions (GC cells cultured under 2% O_2_ or in medium containing CoCl_2_), HIF-1 elevation leads to an increase in the miR-574-5p expression level in GC cells. The authors suggested that the molecular mechanism involves miR-574-5p activating 44/42 kilodaltons (kDa) mitogen-activated protein kinases (MAPKs) by suppressing the expression of its target gene, PTPN3 (a protein tyrosine phosphorylase), thereby promoting angiogenesis by enhancing the expression of VEGF-A [67]. Furthermore, the authors confirmed the role of miR-574-5p in mice tumor xenografts. In this model, the inhibition of miR-574-5p reduced the expression of CD31, a well-known endothelial cell marker [67].

In the second study, Seo et al. reported that another miRNA, miR-210, was found to be progressively upregulated in response to HIFs in hypoxic conditions in GC. In this work, miR-210 was identified as a hypoxia-induced miRNA that plays key roles in biological processes such as cell cycle progression, metabolism, apoptosis, angiogenesis, and metastasis [68].

### 3.3. MicroRNAs Involved in HGF/c-MET Signaling

In this context, we identified one study, carried out by Si et al., in which this signaling pathway has been demonstrated to be involved in GC angiogenesis by miRNA deregulation. In this study, the authors focused on miR-26a/b, which can potentially target HGF in GC. As a result, HGF was upregulated in GC in vivo and in vitro, while miR-26a/b was significantly downregulated. The authors found that the expression of VEGF was induced by HGF, and HGF was upregulated as a result of the downregulation of miR-26a/b. Thus, miR-26a/b appears to promote angiogenesis in GC [69].

### 3.4. MicroRNAs Involved in the PI3K Pathway

In our literature search, we identified 11 relevant studies about miRNAs pertaining to this signaling pathway: four studies describe miRNAs that target PTEN; in three studies, the target was FOXO; three studies reported miRNAs targeting mTOR; and in one study, the target was NF-κB.

#### 3.4.1. MicroRNAs Targeting PTEN

In one study, miR-23a was augmented in HGC-27-derived exosomes, and facilitated angiogenesis by targeting PTEN, as verified by the elevated expression level of VEGF and the reduced expression level of the matricellular protein thrombospondin-1 (TSP-1) [23]. The same study demonstrated that the upregulation of another miRNA, miR-616-3p, in GC tissues resulted in the downregulation of PTEN and PI3K/AKT/mTOR pathway activation through PTEN, which then contributed to EMT and angiogenesis [23]. A second study confirmed the relevance of miR-23a, and a third study confirmed the relevance of miR-616-3p in GC [60,70]. In the study by Du et al., miR-23a was highly expressed in GC tissues, cells, and GC cell-derived exosomes. This miRNA was demonstrated to promote angiogenesis via the repression of PTEN in a co-culture system [70]. The role of miR-616-3p as a promoter of angiogenesis via the PTEN/AKT/mTOR pathway in GC has also been confirmed in the study by Wu et al. [60]. Two other miRNAs, miR-718 and miR-382, have been suggested to target PTEN, thus inhibiting the angiogenesis and progression of GC [71]. In this study by Liu et al., the low expression of PTEN and increased expression of miR-718 were demonstrated to be independent unfavorable prognostic factors for GC [71].

#### 3.4.2. MicroRNAs Targeting FOXO

Two studies reported that miR-155 alone and in GC-derived exosomes could promote angiogenesis in in vitro models of GC by inhibiting FOXO3a expression. MiR-155 also seemed to have the same role in vivo, where it was shown to facilitate angiogenesis in GC [72,73]. Similarly, miR-135b was found to suppress the expression of the FOXO1 protein and enhanced the growth of blood vessels in GC [74].

#### 3.4.3. MicroRNAs Targeting mTOR

The overexpression of miR-18a was shown to inactivate the mTOR pathway and downregulate HIF1α and VEGF expression in a GC cell line named SGC-7901. Furthermore, miR-18a was identified as the cause of the substantial reduction in the number of microvessels in an SGC-7901 xenograft model of GC [63].

Emerging data showed that the circulating miR-17-92 expression level was significantly different between GC individuals and healthy controls, implying that miR-17-92 may be a potential biomarker for GC. In this study, miR-17-92 levels were associated with the progression of advanced GC and the effectiveness of capecitabine and oxaliplatin (CAPOX) chemotherapy [75]. Another work reported that the exogenous expression of miR-101-2, miR-125b-2, and miR-451a decreased the expression of their putative targets mTOR, PIK3CB and TSC1, respectively [76].

#### 3.4.4. MicroRNAs Targeting NF-κB

Finally, a study by Zhang et al. demonstrated that miR-532-5p attenuated NF-κB signaling by directly inhibiting NCF2 expression, while miR-532-5p silencing in GC enhanced NF-κB activity. MiR-532-5p overexpression inhibited GC metastasis and angiogenesis in vitro and in vivo, whereas miR-532-5p silencing had the opposite effect. Furthermore, it was demonstrated that miR-532-5p downregulation was caused by the aberrantly high expression of a long non-coding RNA (lncRNA), LINC01410, in GC [77]. Mechanistically, the overexpression of LINC01410 promoted GC angiogenesis and metastasis by binding to and suppressing miR-532-5p, which resulted in the upregulation of NCF2 and sustained NF-κB pathway activation. Interestingly, NCF2 could, in turn, increase the promoter activity and expression of LINC01410 via NF-κB, thus forming a positive feedback loop driving the malignant behavior of GC [77].

### 3.5. MicroRNAs Involved in STAT-3 Signaling

In our literature search, we identified a study analyzing the effect of miR-874 on the VEGF-A gene in GC. This study demonstrated that the overexpression of this miRNA determined the inhibition of STAT-3 gene expression, leading to the inhibition of VEGF-A expression and, in turn, to a reduction in tumor growth and angiogenesis in vitro and in vivo [78]. These results suggested that miR-874 overexpression may inhibit the VEGF-A pathway, angiogenesis, and tumor growth by acting on the Janus kinase/signal transducers and activators of transcription (JAK/STAT) pathway in GC [62].

**Table 1 biology-10-00146-t001:** MiRNA classification based on referral pathways.

miRNA	Target Genes	Classification	Reference
miR-1	VEGF-A EDN1 MET	VEGF pathway	Azarbarzin et al., 2020 [23]
miR-29a/c	VEGF	VEGF pathway	Zhang et al., 2016 [64]
miR-27b	VEGF-C	VEGF pathway	Liu et al., 2015 [65]
miR-101			
miR-128			
miR-126	VEGF-A	VEGF pathway	Cuzziol et al., 2020 [62] Yang et al., 2015 [63]
miR-590	VEGFR1/2 NRP1	VEGF pathway	Mei et al., 2020 [66]
miR-210	HIF	HIF pathway	Seo et al., 2019 [68]
miR-574-5p	PTPN3	HIF pathway	Azarbarzin et al., 2020 [23] Zhang et al., 2020 [67]
miR-616-3p	PTEN	PI3K pathway VEGF pathway	Wu et al., 2018 [60]
miR-26a/b	HGF	HGF/c-MET signaling	Si et al., 2017 [69]
miR-18a	mTOR	PI3K pathway	Yang et al., 2015 [63]
miR-23a	PTEN	PI3K pathway	Azarbarzin et al., 2020 [23] Du et al., 2020 [70]
miR-101-2	mTOR/PIK3CB/TSC1	PI3K pathway	Riquelme et al., 2016 [76]
miR-125b-2		PI3K pathway	
miR-451a		PI3K pathway	
miR-135b	FOXO1	PI3K pathway	Bai et al., 2019 [74]
miR-382	PTEN	PI3K pathway	Du et al., 2020 [70] Seo et al., 2019 [68]
miR-532-5p	NCF2	PI3K pathway	Zhang et al., 2018 [75]
miR-718	PTEN	PI3K pathway	Du et al., 2020 [70]
miR-155	c-MYB/VEGF FOXO3a	Other—c-MYB PI3K pathway	Azarbarzin et al., 2020 [23] Deng et al., 2020 [72] Zhou et al., 2019 [73]
miR-874	STAT-3/VEGF-A	STAT-3 signaling	Cuzziol et al., 2020 [62] Zhang et al., 2015 [78]

## 4. Discussion

The search for prognostic and predictive biomarkers has been a major objective, together with the development of new treatment options, in oncology in the last few decades. Much investigation has been carried out in order to identify predictive biomarkers of a response to targeted agents and mechanisms of resistance to the same drugs in advanced GC. In this context, several potentially useful biomarkers have been extensively investigated to determine their role in selecting the best candidates for anti-angiogenic therapies in GC. High tissue expression of VEGF-A and -C was found to be correlated with poor prognosis and a higher risk of relapse in patients with resected GC [79,80,81]. The promotion of angiogenesis is typical of chromosome unstable tumors, accounting for the majority (50%) of GC. These tumors typically harbor the overexpression of the gene encoding VEGF-A [82]. On the other hand, although ramucirumab acts by blocking the downstream effects of the VEGF pathway, VEGF subtype expression has never been shown to be a predictor of response to anti-angiogenic treatment in GC. Relevant examples of the highly unsatisfactory results obtained are represented by the biomarker analysis of two placebo-controlled, phase III randomized clinical trials, REGARD and RAINBOW. An exploratory biomarker analysis in the REGARD trial tested VEGFR expression with immunohistochemistry (IHC), while serum samples were assayed for VEGF-C and -D and sVEGFR-1 and -3. The results were not able to identify a strong potentially predictive biomarker of ramucirumab efficacy [83]. Similarly, the biomarker analysis in the RAINBOW study was unsuccessful in identifying circulating predictive factors in plasma samples of GC patients [84]. Ramucirumab is frequently used in combination with paclitaxel because the combination of both drugs has a synergistic inhibitory effect on cell growth [85]. Ramucirumab acts by enhancing the growth inhibition of paclitaxel and the inhibitory effects of chemotherapy on cell migration and actin polymerization. Moreover, the two agents modulate the cell expression of VEGF-A and VEGFR-2 and the signaling of MAPK and the PI3K/AKT/mTOR pathways [85].

Therefore, there is an urgent need to identify innovative biomarkers that are useful to select patients for anti-angiogenic treatment. MiRNAs harbor promising value as predictive and prognostic biomarkers in different types of tumors, including GC [86,87,88]. Indeed, miRNAs are involved in cell cycle regulation and in key processes of tumorigenesis in GC [89,90].

Focusing on angiogenesis, many miRNAs act by interfering with angiogenetic pathways (such as VEGF, HIF, HGF/c-MET, and others) and may play both a prognostic and a predictive role in this setting. Within the VEGF pathway, miR-616-3p was reported to be upregulated in hepatocellular carcinoma and in prostate, lung, and breast cancer [91]. In our analysis, interesting data emerged about this miRNA in GC [60]. Another miRNA involved in the same pathway, miR-126 [92,93], was recognized as a promising therapeutic target in several types of tumors, including GC [94]. MiR-126 is an endothelial-specific miRNA essential for governing vascular integrity and angiogenesis. Previous studies demonstrated that reduced expression of this miRNA may be associated with progression and poor outcomes in lung cancer [95]. Notably, the predictive value of this miRNA has been shown in a study including 68 colorectal cancer patients treated with the anti-angiogenic drug bevacizumab [96]. The authors demonstrated a separation of responders and non-responders in the PFS analysis, with better outcomes for patients with decreased miR-126 levels, (HR: 0.60, *p* = 0.07). Many works reported in our paper suggested a similar potential role for this miRNA [61,62,63]. Considering the HIF pathway, miR-210 is of particular relevance. This miRNA has been demonstrated to be differentially expressed in healthy controls and in patients affected by lung cancer [97]. Other evidence suggested a prognostic role for miR-210 in metastatic renal cell carcinoma [98], as well as in osteosarcoma [99]. Furthermore, considering the PI3K signaling pathway, we identified 11 relevant studies about miRNAs with promising value as predictive or prognostic biomarkers in GC. Of these, miR-18a was identified as a promising prognostic marker in lung and bladder cancer [100]. This miRNA has also been evaluated as a potential diagnostic marker for GC [63]. Similar results were reported for colorectal cancer [101]. Belonging to the same pathway, miR-17-92 showed a predictive role with respect to capecitabine and oxaliplatin chemotherapy in GC patients [75]. Notably, the prognostic and predictive values of this miRNA in rectal cancer patients has also been described [102]. Finally, miR-874, related to the STAT-3 pathway, has emerged as a potentially useful diagnostic biomarker for colorectal cancer (CRC) [103]. In this study, miR-874 levels were significantly downregulated in CRC patients compared to healthy controls. Similarly, miR-874 has been recognized as playing a tumor suppressor role in non-small cell lung cancer (NSCLC) both in vitro and in vivo [104]. Similar results have been achieved in pancreatic cancer [105].

Therefore, the data collected and analyzed in this review suggest that different angiogenesis-related miRNAs could potentially represent innovative diagnostic and prognostic biomarkers for advanced GC patients. Additionally, some of these miRNAs may be correlated with benefit and toxicity to anti-angiogenic agents in this group of patients, even though these data have not been validated prospectively.

## 5. Conclusions

MiRNAs are involved in the regulation of neoplastic angiogenesis, which is a crucial mechanism in GC onset and progression.

In future studies, miRNAs should be evaluated as candidate biomarkers with prognostic and predictive features, especially in the majority of GC cases that harbor a pro-angiogenic molecular signature. High data throughput techniques such as NGS may be used in sub-analyses of large clinical trials, both on available tissue samples and liquid biopsies. In this way, the identified miRNAs may be associated with survival outcomes and clinical response to therapies.

## Figures and Tables

**Figure 1 biology-10-00146-f001:**
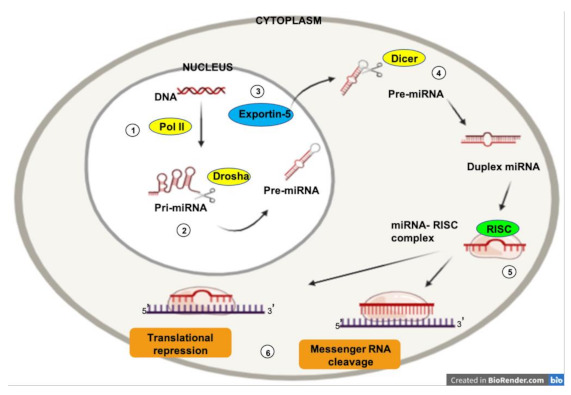
The expression of micro RNAs (miRNAs) is a complex biological process. RNA polymerase II is responsible for the transcription of a precursor RNA, several thousand nucleotides long and with a loop (hairpin) shape, known as “primary miRNA” (pri-miRNA) (1). The nuclear endonuclease Drosha processes the pri-miRNAs by cleaving the distal portion and making shorter chains (70–100 nucleotides) (2). This yields the pre-miRNA, which is transported into the cytoplasm via the nuclear receptor exportin-5 (3). Once in the cytoplasm, the Dicer enzyme processes the pre-miRNA to obtain a short (19–25 nucleotides) double chain RNA sequence (4). Subsequently, one of the two chains is rapidly degraded, while the remainder represents the mature miRNA. Once processed, mature miRNAs can interact with Ago2, an enzyme of the Argonaute family of endonucleases, to form so-called RNA-induced silencing complexes (RISCs) (5). This allows the interaction between the mature miRNA and the target mRNA (6). The miRNA can then perform its function (degradation or translational repression).

**Figure 2 biology-10-00146-f002:**
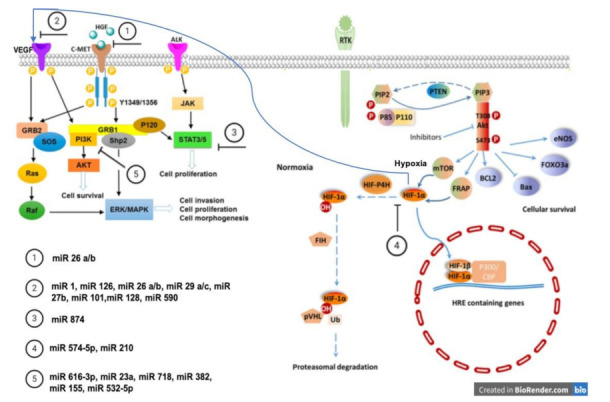
Major groups of micro RNAs (miRNAs) and their related pathways by which they act on endothelial cells. In normoxia, HIF-1α is hydroxylated and ubiquinated. In hypoxia, hypoxia inducible factor-1α (HIF-1α) promotes angiogenesis by enhancing the expression of vascular endothelial growth factor-A (VEGF-A). MiRNAs act on the VEGF pathway and hepatocyte growth factor (HGF)/ tyrosine-protein kinase MET (c-MET), phosphoinositide 3-kinase (PI3K), signal transducer and activation of transcription-3 (STAT-3) and HIF signaling.

## Data Availability

Not applicable.

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
