# Peer review of "The Role and Expression of Angiogenesis-Related miRNAs in Gastric Cancer"

_biology, 2021, doi:10.3390/biology10020146_

Round 1

Reviewer 1 Report

This is a very interesting review providing an in-depth summary on the role of miRNAs as post-transcriptional regulators in angiogenesis. The authors provide a coherent overview of the role of miRNAs involved in several angiogenesis pathways related to Gastric Cancer.

Major comments:

Although the authors mention on page 1 (abstract) that they have evaluated the prognostic and predictive role of angiogenesis miRNAs in GC it is not clear to me how this is achieved in this review. This is further re-iterated in page 4 lines 135-127.Apart from a very detailed background on miRNAs involved in angiogenesis pathways in gastric cancer this review does not demonstrate why and how these can be utilised in such manner. Could the authors please explain how they achieve this aim?

The discussion does not either give any such perspective. Instead, the first half focuses on the lack of discovery of predictive biomarkers to ramucirumab in REGARD and RAINBOW  biomarker exploratory analyses studies and the second on the development of new antiangiogenic  drugs in GC. On an unconnected manner.

It would be useful if the authors could provide some examples from the existing literature on the prognostic and predictive ability of miRNAs in gastric cancer to give some perspective towards this direction.

I would also recommend that the authors rephrase their statement on page 1 lines 36-3.

Page 2 , line 59 please replace “multi-aspect” with multiple aspects

Page 2, lines 87-89: please briefly describe whether there has been any (preliminary) evidence of efficacy of the mentioned TKIs and anti-EGFR antibodies. It is advisable to also expand the references of this section.  Lines 414-433 of page 12 (discussion) would be better suited in this section than discussion.

Page 3, lines 99-100:  could the authors specify which angiogenesis inhibitors they refer to?

Page 3, line 104: please rephrase this sentence. one suggestion could be :  “to reduce unnecessary toxicity while maximising clinical benefit”

Page 3, line 108: please replace  “MiRNA” with “ MiRNAs”

Page 4 line 146, please change “concerned” to “regarding”

Page 5, line 164, please remove “the”

Page 5, line 178, please replace “resulted” with “was”

Page 5, line 179: please replace “cells lines” with “cell lines”

Page 7, line 285 : This sentence seems unrelated to the previous and later content. it would be more relevant in the beginning of the paragraph where authors describe the pi3k pathway.

Page 8, line 300, Which studied do the authors refer to?

Page 8, line 303 the authors have used past tense when reported some studies an present tense for others. For consistency, please use past tense for all.

Page 12 , line 408 : please replace instable with unstable

Author Response

Dear Reviewer, thanks for your useful hints. Please see below the answers to your kind comments.

Major comments:

- Although the authors mention on page 1 (abstract) that they have evaluated the prognostic and predictive role of angiogenesis miRNAs in GC it is not clear to me how this is achieved in this review. This is further re-iterated in page 4 lines 135-127.Apart from a very detailed background on miRNAs involved in angiogenesis pathways in gastric cancer this review does not demonstrate why and how these can be utilised in such manner. Could the authors please explain how they achieve this aim?

We modified our abstract according to your comment: "In our review we aimed at analyzing the available data on the role of angiogenesis-related miRNAs in GC."

- The discussion does not either give any such perspective. Instead, the first half focuses on the lack of discovery of predictive biomarkers to ramucirumab in REGARD and RAINBOW  biomarker exploratory analyses studies and the second on the development of new antiangiogenic  drugs in GC. On an unconnected manner. It would be useful if the authors could provide some examples from the existing literature on the prognostic and predictive ability of miRNAs in gastric cancer to give some perspective towards this direction.

We modified all the discussion considering your comments.

- I would also recommend that the authors rephrase their statement on page 1 lines 36-3.

Rephrased as it follows: "In our review we aimed at analyzing the available data on the role of angiogenesis-related miRNAs in GC."

- Page 2 , line 59 please replace “multi-aspect” with multiple aspects

Replaced as suggested.

- Page 2, lines 87-89: please briefly describe whether there has been any (preliminary) evidence of efficacy of the mentioned TKIs and anti-EGFR antibodies. It is advisable to also expand the references of this section.  Lines 414-433 of page 12 (discussion) would be better suited in this section than discussion.

We modified and replaced this section, now in chapter 1.2 (Angiogenesis and Gastric Cancer).

- Page 3, lines 99-100:  could the authors specify which angiogenesis inhibitors they refer to?

We replaced all the part concerning angiogenesis inhibitors as reported above. We refer to all the biological agents reported in chapter 1.2.

- Page 3, line 104: please rephrase this sentence. one suggestion could be :  “to reduce unnecessary toxicity while maximising clinical benefit”

Rephrased as suggested.

- Page 3, line 108: please replace  “MiRNA” with “ MiRNAs”

Replaced as suggested.

- Page 4 line 146, please change “concerned” to “regarding”

Replaced as suggested.

- Page 5, line 164, please remove “the”

Removed as suggested.

- Page 5, line 178, please replace “resulted” with “was”

Replaced as suggested.

- Page 5, line 179: please replace “cells lines” with “cell lines”

Replaced as suggested.

- Page 7, line 285 : This sentence seems unrelated to the previous and later content. it would be more relevant in the beginning of the paragraph where authors describe the pi3k pathway.

All the pathways description have been moved to chapter 1.2 (Angiogenesis and Gastric Cancer).

- Page 8, line 300, Which studied do the authors refer to?

All the section was modified, now it is clear which studies we refer to.

- Page 8, line 303 the authors have used past tense when reported some studies an present tense for others. For consistency, please use past tense for all.

Verbs replaced as suggested.

- Page 12 , line 408 : please replace instable with unstable

Replaced as suggested.

Reviewer 2 Report

The authors of the current review have be congratulated for their effort.

It is a very good paper with nice review of all the available literature.

I have two comments to make which in my opinion have to be taken into consideration prior accepting the paper for publication.

The first comment is related to the first section of the review. In my opinion a paragraph regarding methods of detecting and measurement of miRNAs would be very helpful to clinicians who are not familiar with the laboratory methods used to detect miRNA.

The second relates to the section "Discussion". In my opinion is not well balanced as too little is written regarding miRNAs. I would suggest maybe to discuss difference between miRNA in gastric cancer and other types of cancer. Also maybe authors could compare results of their review with older reviews in the specific scientific field.

Thank you.

Author Response

Dear Reviewer, thanks for your useful hints. Please see below the answers to your kind comments.

I have two comments to make which in my opinion have to be taken into consideration prior accepting the paper for publication.

- The first comment is related to the first section of the review. In my opinion a paragraph regarding methods of detecting and measurement of miRNAs would be very helpful to clinicians who are not familiar with the laboratory methods used to detect miRNA.

This was a good point, thanks for your hint. We included a new section on methods of detecting in chapter 1.3 "MicroRNAs and Cancer".

- The second relates to the section "Discussion". In my opinion is not well balanced as too little is written regarding miRNAs. I would suggest maybe to discuss difference between miRNA in gastric cancer and other types of cancer. Also maybe authors could compare results of their review with older reviews in the specific scientific field.

This was also a good point. We discussed differences between miRNAs in GC and other types of cancer. Discussion was significantly modified.

Reviewer 3 Report

In this article entitled " Role and Expression of Angiogenesis-Related miRNAs in 2 Gastric Cancer" Giuppi et al reviewed the role of microRNAs in influencing several pathways involved in angiogenesis and gastric cancer.

While this article has good scientific merit, I think it needs to be re-arranged and more focused.

The “Introduction” section should be arranged:

  1. Remove/ compile generic information about cancer, angiogenesis etc all these broad discussions deviating the focus of the review.
  2. Introduce the pathways involved angiogenesis and gastric cancer, why they are important
  3. Then you can dive into regulation of these pathways by microRNA
  4. You should draw a connection between pathways involved in angiogenesis and cancer how they overlap and where microRNA fits in that image
  5. Several missing references against statements
  6. Future direction is important for a review that gives direction to the field. Make it a  separate paragraph and share your views.
  7. Make the images clear and easy to follow for the readers.

Author Response

Dear Reviewer, thanks for your useful hints. Please see below the answers to your kind comments.

The “Introduction” section should be arranged:

- Remove/ compile generic information about cancer, angiogenesis etc all these broad discussions deviating the focus of the review.

Section removed as suggested.

- Introduce the pathways involved angiogenesis and gastric cancer, why they are important

Please find chapters 1.1 ("Angiogenesis and cancer") and 1.2 ("Angiogenesis and gastric cancer") that were totally rewritten as suggested.

- Then you can dive into regulation of these pathways by microRNA

Pathways description and involved microRNAs description were divided as suggested.

- You should draw a connection between pathways involved in angiogenesis and cancer how they overlap and where microRNA fits in that image

Figure 2 was modified as suggested. We drew and explained the main connection between the HIF and VEGF pathway.

- Several missing references against statements

All missing references were included.

- Future direction is important for a review that gives direction to the field. Make it a separate paragraph and share your views.

We included a final paragraph in the conclusions section.

Make the images clear and easy to follow for the readers.

We added figure legends. 

Round 2

Reviewer 1 Report

Thank you for taking into account my comments.

Author Response

Thanks a lot for your kind revision.

Reviewer 2 Report

In my opinion your review article has now been further improved. I would only delete the phase "Neither..... so far." in lines 436-437 in page 13 as I do not think it fits well in conjunction with the other phrases in the section "Conclusions". After that I would accept the paper.

Kind regards

Author Response

Thanks a lot for your kind revision. We modified the text according to your suggestion.

Reviewer 3 Report

Authors have improved the quality of the article significantly. I have no more concern.

Author Response

Thanks a lot for your kind revision.